# Psychological Capital and Self-Acceptance Modified the Association of Depressive Tendency with Self-Rated Health of College Students in China during the COVID-19 Pandemic

**DOI:** 10.3390/bs13070552

**Published:** 2023-07-03

**Authors:** Yongcheng Yao, Junyan Yao, Shuyan Chen, Xiaohui Zhang, Hongling Meng, Yuping Li, Lingeng Lu

**Affiliations:** 1School of Mathematics and Statistics, Zhengzhou Normal University, Zhengzhou 450044, Chinahnmhl@126.com (H.M.); liyuping970301@sina.com (Y.L.); 2College of Elementary Education, Capital Normal University, Beijing 100048, China; 3Department of Chronic Disease Epidemiology, Yale School of Public Health, Yale University, New Haven, CT 06520-8034, USA; lingeng.lu@yale.edu; 4School of Medicine, Yale University, New Haven, CT 06520-8034, USA

**Keywords:** college students, self-rated health, depressive tendency, psychological capital, self-acceptance

## Abstract

Background: To explore the association between the self-reported health status, depressive tendency, psychological capital, and self-acceptance of college students in China during the COVID-19 pandemic. Methods: Using the online survey platform “questionnaire star”, a two-phase cross-sectional study was conducted on a total number of 1438 undergraduates with informed consents. The questionnaires of Self-Rated Health Measurement Scale (SRHMS), the Center for Epidemiological Studies-Depression Scale (CES-D), Psychological Capital Questionnaire (PCQ-24), and self-acceptance questionnaire were administered to each participant. Results: Male college students had significantly higher depressive tendency scores than female (17.59 vs. 15.82) (*p* < 0.01). College students having no siblings had significantly higher psychological capital scores than those having siblings (108.63 vs. 105.60) (*p* < 0.05). Exercise had significantly positive associations with self-rated health, psychological capital, and self-acceptance scores, while online time per day had significantly negative associations. Multivariate analysis showed that the interaction between depressive tendency, psychological capital, and self-acceptance was statistically significant (*β =* 0.004, *p =* 0.013 for phase 1 and *β =* 0.002, *p =* 0.025 for phase 2) in health status with depressive tendency ranking the top (*β =* −0.54 for phase 1 and −0.41 for phase 2, *p* < 0.001). Mediation analysis showed that psychological capital and self-acceptance modified the association of depressive tendency with health status. Conclusion: Physical exercise is beneficial to both physical and psychological health. Depressive tendency is the main risk factor that associates with self-rated health. Regardless of depressive tendency level, high psychological capital and self-acceptance could improve college students’ health.

## 1. Background

Severe acute respiratory syndrome coronavirus 2 (SARS-CoV-2) is a novel coronavirus, causing highly contagious coronavirus disease (COVID) [1,2]. Since it emerged in December 2019 in Wuhan, China [3,4], there have been over 605 million confirmed cases of COVID-19, and over 6.4 million deaths worldwide as of 12 September 2022. Epidemiological studies show that cases in other parts of the world had direct travel history from affected areas or with exposure or contact to a known case [5,6]. To help slow the spread of COVID-19, many local governments in many countries including China later on took measures to restrict traveling, indoor gatherings, business operations, and enforce social distancing. To meet the measurement for containing COVID-19, the Ministry of Education of China issued mandatory requirements, closing all schools for the Spring semester until reopen, and all students were asked to stay at home and opt for virtual online courses rather than in-person instruction on campus. This physical isolation measurement significantly flattened the disease curve, however, it also led to a variety of psychological adverse effects, e.g., anxiety, fear, grief, depression [7], and other serious health problems (e.g., obsessive-compulsive disorder (OCD) [8], due to fewer opportunities to engage with peers and/or excessive online surfing for communication, fear of future, and change in academics). Mental health issues, such as difficulty sleeping or eating, abuse of alcohol consumption or substances, and worsening chronic conditions, arose due to worry and stress over the pandemic [2,9,10,11]. The prevalence of depression symptoms significantly increased by more than 3-fold during COVID-19 than before the COVID-19 pandemic in United States adults [12]. Over 50% of health care workers in China who took care of patients with COVID-19 reported depression symptoms [13]. Reduced sleeping time in health care worker accelerated anxiety and stress [14]. Elevated prevalence of anxiety was demonstrated in a study of Chinese medical students [15]. A recent study reported that 21.1% undergraduate and postgraduate college students (aging 18 years and older) from 152 countries reported depression [16]. In Shanghai, depressive symptoms dramatically increased in college students with the introduction of quarantine in periods 2 and 3 compared to period 1 [17]. A similar observation was found in a study conducted in Poland, with elevated stress and lower general health in University students during the pandemic [18]. A recent study conducted in UK young people (aged 16–24 years) showed a positive association between the COVID-19 pandemic and anxiety and distress [19]. The COVID-19-related shutdown negatively affected university students’ mental and physical health [20]. During the COVID-19 pandemic, young people aged 18 to 29 was the most affected group, with 42% reporting anxiety and 36% depression, followed by the 30-to-39-year-old group, with 34% reporting anxiety and 28% depression, whereas older people were far less likely to report these symptoms [21]. In China, 35% of medical students reported depression, and a higher prevalence of depression in college students from rural families was observed than those from urban families [7]. Even as high as 70% of graduate students reported anxiety, depression, and stress in a recent study conducted in the United States [22]. A meta-analysis of international population-based studies showed 31.9% of individuals suffering anxiety and 33.7% suffering depression during the COVID-19 pandemic [23].

Depression is a mental disorder that is characterized by loss of interest and enjoyment and reduced energy, as well as difficulties in decision-making. Severe depression can lead to suicide, accounting for up to 60% of suicides [24], which is the third leading cause of death among young people in the US, and is a significant public health problem worldwide [25]. To reduce COVID-19 pandemic-induced stress, a series of shareable resources on coping with COVID-19 were provided, educating people on how to fight against COVID-19-related stress with a positive attitude, which may buffer the pandemic-induced mental and physical health consequences [20]. Although young college students are relatively less susceptible to severe COVID-19 in comparison to the elderly (greater than 65 years old), campus closure, home-confinement, and online classes may still bring negative impacts on their mental and physical health, which may persist even when society is re-opened [20,26]. Thus, it is a priority to assess college students’ health and assess factors influencing mental health in the context of COVID-19, providing a theoretic basis to make strategies or prepare additional campus resources on how to keep students healthy in the context of COVID-19.

Self-acceptance is an important part of self-awareness, which means that an individual can affirm the positive values in all aspects of his/her own body, ability, and morality, and accept all facets of his/her own reality [27]. Low self-acceptance easily induces disappointment, shame, worthlessness, and other severe negative emotions, e.g., depression [28]. In contrast, high self-acceptance could promote health by reducing mortality [29]. More satisfaction in the use of e-learning (remote online learning) was positively associated with reduced stress, anxiety, and depression in Italian nursing students during the COVID-19 pandemic [30]. Meaning in life was negatively associated with depression and anxiety in undergraduate students during the COVID-19 pandemic [31]. These observations indicate that self-acceptance may affect COVID-19-related depression.

There are four aspects in psychological capital: self-efficacy (confidence), optimism, hope, and resilience. Self-efficacy makes an individual take effort with confidence to succeed at challenging tasks. When facing difficulties, one can persevere toward goals with a positive attribution and expectation, redirecting paths to goals in order to achieve them (optimism and hope). Even beset by problems and failure, one can bounce back from the adversity to attain success (resilience). It has been suggested that psychological capital could ameliorate the negative consequences of stress on mental health, and enhance positive psychological outcome in undergraduates [32]. Plenty of psychological capital empowers individuals with resilience and strength to challenge adverse situations, protecting social workers from burnout and traumatic stress [33,34]. High psychological resilience makes individuals enable to “bounce back” from negative emotional experiences in confronting stressful conditions, protecting nurses from burnout [35,36]. Increasing psychological well-being was associated with reduced anxiety, depression, and stress in university students during the COVID-19 pandemic [37]. An association between social satisfaction and better mental health was also found in Bangladesh university students during the pandemic [38].

In a recent survey study conducted in UK, it has been shown that worsening mental health due to the COVID-19 pandemic was reported in the majority (55%) of participants, which was associated with female gender, young and unemployed/student status [39]. In the United States, as high as 95.7% of the college students had mood disorders, showing elevated fear and stress, leading to decreased learning quality during the COVID-19 pandemic [40]. Taken together, we hypothesize that there are associations between depressive tendency, psychological capital, self-acceptance, and self-rated health, and that the interaction exists between depressive tendency, psychological capital, and self-acceptance in health in college students during the COVID-19 pandemic in China. Thus, the purposes of this study were to evaluate the mental health status in Chinese college students and to investigate the associations between depression, psychological capital, self-acceptance, and self-rated health during the COVID-19 pandemic, providing a theoretic basis to make strategies to keep students healthy in the context of COVID-19.

## 2. Methods

### 2.1. Participants

#### Sample Size and Sampling Technique

The sample size (n) was calculated based on a single population proportion formula of N = [u_α_^2^π(1 − π)]/δ^2^, assuming δ = 0.03, α = 0.05, u_α_ = 1.96. A previous meta-analysis study showed that the prevalence of depression was 39% among college students during the COVID-19 pandemic [41], and that is π = 0.39. Thus, we estimated that at least 1016 college students were needed.

A two-phase cross-sectional study was conducted with phase 1 in May and phase 2 in December 2020 on college students from an undergraduate college in Henan Province who were chosen using a cluster sampling approach. All participants provided informed consent in this study. All procedures performed in this study involving the participants were approved by the institutional research committee of Zhengzhou Normal University, and we in accordance with the 1964 Helsinki declaration and its later amendments or comparable ethical standards. Overall, 367 valid questionnaires at phase 1 and 1071 at phase 2 were collected via an online system of the questionnaire star survey platform. Of them, 507 (35.5%) were boys and 931 (64.7%) were girls. Participants were 17 to 24 years old, with an average age of 20 years. The questionnaire includes general information of demographics for each participant, e.g., sex, age, siblings, physical exercise (above half an hour each time), and Internet surfing duration every day. Informed consent was obtained from all participants included in the study.

**Health status:** The self-rated health measurement scale (SRHMS) [42] was used to evaluate the health status of college students. The scale includes 48 items in three dimensions: physical, mental, and social health. Using a 0 to 10 grade scoring method for each item, the score of each dimension was the sum of the scores of all items in the dimension. A higher score represents a better health status. The Cronbach’s α coefficient of the scale in this study was 0.939.

**Depressive Tendency:** Depressive tendency was assessed using the questionnaire of “Center for Epidemiologic Studies-Depression Scale” (also known as the “Self-rating Depression Scale for Streaming Calls”) (CES-D) [43]. The scale contains four dimensions of somatization symptoms, depression, positive emotions, and interpersonal problems, with a total of 20 items. Each item uses a Likert scale from 0 (never or rarely) to 3 (most of the time or all of the time). A summed score ranges from 0 to 60, whereby a higher score represents an increased level of depressive tendency. In this study, the scale Cronbach’s α was 0.914.

**Psychological capital:** Psychological capital was assessed using the Psychological Capital Questionnaire (PCQ), which was developed by Luthans et al., and adopted to the version in Chinese by Li et al. [44]. The questionnaire contains 24 items measuring four dimensions of self-efficacy, hope, resilience, and optimism. A Likert 6-level scoring was used from 1 (“strongly disagree”) to 6 (“strongly agree”). Higher scores represent more positive psychology for individuals, and the more resources they can use to improve their performance on the job and their success. The Cronbach’s α coefficient of the scale in this study was 0.958.

**Self-acceptance:** The self-acceptance questionnaire, which includes two factors of self-acceptance and self-evaluation, was compiled by Cong and Gao in 1999 [45]. There are 16 items with each item scoring 1–4. The summed score ranges from 16 to 64 points for two factors, and each factor scores from 8 to 32 points. A higher score represents a higher degree of self-acceptance. The Cronbach’s α coefficient of the questionnaire in this study was 0.834.

### 2.2. Statistical Analysis

All statistical analyses were performed using SPSS 18.0 software. Online surfing time was grouped based on the cutoff of 3 and 6 h as previously described elsewhere [46]. A normality test (the absolute values of Skew and Kurtosis are both less than 1) was applied to check the distribution of the numeric variables first to choose appropriate statistics and statistical analysis methods. Numerical variables are presented as mean ± standard deviation (SD). Either t-test or Analysis of Variance (ANOVA) was used to analyze the differences between the groups. The Bonferroni method was used for the correction of multiple comparisons, and the Dunnett^,^ sT3 method was used when the variance was uneven. Pearson correlation analysis was carried out on college students’ self-rated health, psychological capital, self-acceptance, and depressive tendency. Multiple regression analysis was used to explore the ternary interaction of psychological capital, self-acceptance, and depressive tendency on self-rated health, and the interactive relationship diagram was constructed according Mod-Graph [47]. The PROCESS plugin for SPSS was used to assess the strength of the indirect effects in mediation analyses [48]. In PROCESS, total, direct, and indirect effects were calculated and tested for significance [49,50,51]. The mean value for the ab (a × b) product across the bootstrapped samples provided a point estimate of the indirect effect and 95% confidence intervals (CI). Each analysis was based on 5000 bootstrapped samples, as suggested by Preacher [50]. *p* < 0.05 was considered as statistically significant.

## 3. Results

### 3.1. Common-Method Variation Test

Harman’s single factor method was performed to analyze the common-method variance of variables. The eigenvalues of 17 factors were >1, with the first factor explaining 26.81% of the variance. There was no severe common-method variance in this study based on the critical threshold of 40% [52].

### 3.2. Differences in Self-Rated Health, Psychological Capital, Depressive Tendency, and Self-Acceptance among Demographics and Lifestyles

Male college students had significantly higher depressive tendency scores than female (17.59 vs. 15.82) (*p* < 0.01) (Table 1). College students having no siblings had significantly higher psychological capital scores than those having siblings (108.63 vs. 105.60) (*p* < 0.05). Exercise significantly improved self-rated health, psychological capital, and self-acceptance (*p* <0.001). College students who exercised more than once a week had significantly higher scores of self-rated health, psychological capital, and self-acceptance than college students who exercised once a week and who did not exercise (*p* < 0.05). The scores of health self-assessment, psychological capital, and self-acceptance of college students who exercise once a week are significantly higher than those who do not exercise (*p* < 0.05). There were significant differences in the scores of self-tested health, psychological capital, and self-acceptance between college students who spent different hours on the Internet per day (*p* < 0.001). The students who spent less than 3 h per day online had significantly higher scores of self-rated health (340.47 vs. 323.24), psychological capital (108.48 vs. 100.79), and self-acceptance (42.46 vs. 39.70) than those who spent more than 6 h per day.

### 3.3. Correlation between Self-Rated Health, Depressive Tendency, Psychological Capital, and Self-Acceptance

Pearson correlation analysis results are illustrated in Table 2. We found a significantly negative correlation of depressive tendency with either self-rated health (correlation coefficient = −0.63, *p* < 0.01), or psychological capital (correlation coefficient = −0.46, *p* < 0.01), or self-acceptance (correlation coefficient = −0.56, *p* < 0.01), respectively. There were significantly positive correlations between psychological capital and self-acceptance (correlation coefficient = 0.51, *p* < 0.01), and between self-rated health and either psychological capital (correlation coefficient = 0.56, *p* < 0.01) or self-acceptance (correlation coefficient = 0.52, *p* < 0.01).

### 3.4. Interaction between Depressive Tendency, Psychological Capital, and Self-Acceptance in Self-Rated Health

To further analyze the effect of depressive tendency, psychological capital, and self-acceptance on and their interaction in self-rated health, we performed three different multiple regression models for each phase. The results of phase 1 were validated in phase 2, and the results are demonstrated as in Table 3. In model 1 in which the main effect of three variables only was included, we found that depressive tendency had a significantly negative effect on self-rated health with the coefficient of −0.53 for phase 1 and −0.40 for phase 2 (*p* < 0.001), whereas both psychological capital and self-acceptance had a significantly positive effect, and their coefficients were 0.27 (*p* < 0.001) and 0.11 (*p* = 0.011) vs 0.30 (*p* < 0.001) and 0.13 (*p* < 0.001), respectively. The main effect of the three variables could explain 62.6% and 47.2% of the variation of self-rated health. In model 2, we added the two-way interaction items of psychological capital, self-acceptance, and depressive tendency besides the main effects. The significance for the three main effects still remained, however no significance was found for all three two-way interaction items (*p >* 0.05). In model 3, we included one three-way interaction of psychological capital, self-acceptance, and depressive tendency beyond model 2. We found that the three-way interaction was statistically significant (*β* = 0.004, *p* = 0.013, and *β* = 0.002, *p* = 0.025). The main effect of depressive tendency was the greatest with a negative coefficient (*β* = −0.54 and −0.41, *p* < 0.001).

The Bootstrap method was applied in mediation analysis to further investigate the mediating effect of psychological capital and self-acceptance on depressive tendency and self-rated health. The results are shown in Table 4. We found statistically significant indirect effects for both psychological capital and self-acceptance, which accounted for 24.7% and 20.6%, respectively.

In order to better visualize the interaction of depressive tendency, psychological capital, and self-acceptance in self-rated health of all the college students, we further constructed a three-way interaction diagram based on the method described by Dawson and Richter [47] (Figure 1). With the score of depressive tendency increasing, the significantly decreased score of self-rated health was observed regardless of psychological capital and self-acceptance. The slope (declining rate) for the students with high psychological capital and high self-acceptance was significantly smaller than that for those with high psychological capital and low self-acceptance (*p* = 0.041). Moreover, at the same level of depressive tendency, students with high psychological capital and high self-acceptance had the best self-rated health status, whereas the status of self-rated health for those with low psychological capital and low self-acceptance was the worst.

## 4. Discussion

In this study, we demonstrated the associations between depressive tendency, psychological capital, self-acceptance, and self-rated health in college students in China during the period of the COVID-19 pandemic. We found that depressive tendency had a significantly negative association with self-rated health, whereas both high psychological capital and high self-acceptance significantly improved self-rated health. We also found that there was a statistically significant three-way interaction. Psychological capital and self-acceptance modified the association of depressive tendency with self-rated health. The association strength of depressive tendency with health was significantly alleviated when college students had high psychological capital and self-acceptance in comparison to those with either low psychological capital or low self-acceptance. To our knowledge, this is the first study to investigate the interaction of depressive tendency, psychological capital, and self-acceptance in college student health during the pandemic. In addition, we also found that students who came from a one-child family (no sibling) had a higher score of psychological capital. One possibility for this phenomenon is that they probably start to play together and make friends with peers from other families earlier, and through such an interaction they can learn mutual and social relationship and obtain support and warm from friends.

The COVID-19 pandemic worldwide directly or indirectly affects mental health. Individuals infected by COVID-19 experience post-traumatic stress syndrome, depression, and anxiety [4]. Some may suffer multiregional cognitive dysfunction, persistent headache, and extreme fatigue with sleep disorders [4]. Fear to contract highly contagious COVID-19 with severe sequalae and complications afflicts more individuals. Over 90% of university students in the United States reported the mood disorders of stress and anxiety with decreased learning quality during the COVID-19 pandemic [40]. Worsening mental health was demonstrated in 55% of the participants during the period of the COVID-19 pandemic [39]. There were associations between mental health and female, young, and unemployed/student status [39]. It has been shown that during the COVID-19 pandemic, young people aged 18 to 29 years was the most affected group with anxiety and depression, followed by people aged 30 to 39 years and older people [21]. Approximately one third of medical students reported depression in China [7], as well as worldwide during the COVID-19 pandemic [23]. In our study, we found that the average score of depressive tendency was relatively higher than 16 in each category of the variables, and the intensity had been enough to seek medical counseling, although there were no significant differences found between the groups. This finding suggests that the majority of Chinese college students had depression, and it is in agreement with other studies, in which almost half of the youth aged 18–29 years old reported depression during the pandemic [21]. Depression tendency is the major negative factor in self-rated health of college students. Thus, this health issue in college students should not be ignored; education administrators should pay more attention to it and take measurement for psychological intervention, effectively reducing students’ anxiety, and depression during the pandemic. Given that physical exercise could alleviate depression symptoms, college students staying at home for isolation and social distancing should be encouraged to maintain a certain amount of exercise rather than sitting at home all day, although in our study, higher but not statistically significant scores were found in college students who had no exercise or spent more time using the Internet. This non-significant difference may be due to the too widely spread and highly contagious disease, and very restricted confining measurement carried out in China. However, exercise clearly benefits self-rated health, psychological capital, and self-acceptance, whereas long-time online surfing does bring negative effects on health, psychological capital, and self-acceptance. These findings are in line with the previous reports [53,54,55]. Physical activity is not only beneficial for physical health, but also can alleviate psychological symptoms. It has been reported that exercise intervention has a positive effect on the physical and mental health of college students [56].

High score of self-acceptance indicates that individuals can accept themselves and understand themselves appropriately when encountering various situations in reality. With self-knowledge, they maintain understanding towards themselves and their own specialty [45]. Individuals with a clear self-awareness usually can make relatively correct evaluations in the face of difficulties and challenges, and can finally overcome difficulties and achieve success in combination with a positive mental state. In our study, we found that when the intensity of depression is too high, even college students who have plenty of resources for help and high self-acceptance can still have their health impacted. The effect is, however, relatively weaker in those who have more psychological capital and higher self-acceptance than those with less psychological capital or lower self-acceptance. This is consistent with the previous findings that high self-acceptance and psychological capital can effectively alleviate the negative effect of depression on health [57,58]. Manuli and colleagues reported that nurse school students who satisfied e-learning due to the COVID-19 prevention measures had less stress, anxiety, and depression [30]. Another study showed that undergraduate students who felt a meaning in life had less depression and anxiety during the COVID-19 pandemic [31]. Moreover, people with plenty of psychological capital generally believe that they have abilities and more resources to apply, thereby overcoming difficulties and achieving success when facing challenges, and showing a more determined attitude [59]. There is a positive association between high psychological resilience and low risk of burnout and low job performance [60]. Therefore, self-acceptance and psychological capital are positively associated with health [54,58].

There exist some limitations in this study. As other cross-sectional studies, we cannot make causal inference based on the association between the factors we investigated and self-rated health. Secondly, the participants voluntarily took part in the survey, bias might exist, and the participants may not be representative of the whole population of college students, particularly when the sample size is relatively small. However, this is a two-phase study, and the findings from two phases are quite consistent, suggesting that the results are quite robust, and warranting further studies with a relatively large sample size. Longitudinal or intervention studies can be carried out for casual inference. It would also be interesting to investigate the association of individual aspects of psychological capital each such as resilience with depression tendency in college students to understand possible differences or similarity among the four different aspects in psychological capital during the COVID-19 pandemic.

## 5. Conclusions

In summary, we demonstrated the associations of depressive tendency, psychological capital, and self-acceptance with health in college students in China during the period of COVID-19 pandemic in this two-phase study. We found that depressive tendency was a major risk factor negatively associating health, and psychological capital and self-acceptance modified the association of depressive tendency on health in college students. The association observed in phase 1 was validated in phase 2 data, suggesting the results are quite robust. This study provides knowledge for our educational administrators to make strategies for the improvement and maintenance of college students’ health, particularly mental health during the period of the COVID-19 pandemic. Although the end of the COVID-19 pandemic has been announced, post-effects on mental health may not vanish immediately. In addition, humanity may from time to time suffer from such a pandemic in the future, and such studies could have useful insight in these contexts.

## Figures and Tables

**Figure 1 behavsci-13-00552-f001:**
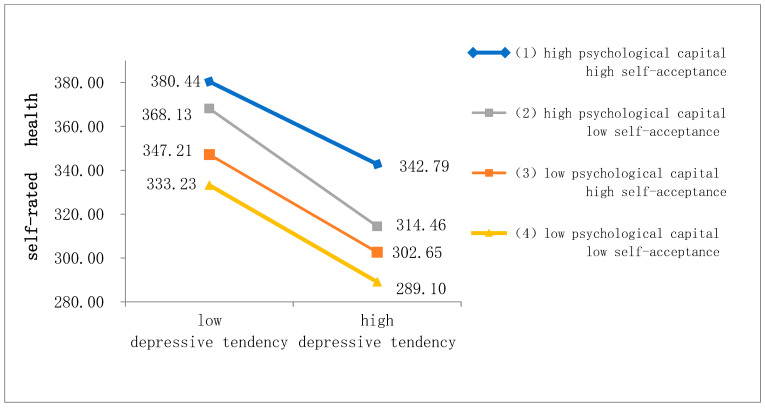
Three-way interaction of psychological capital, depressive tendency, and self-acceptance in self-rated health. Blue line represents the individuals with high psychological capital and self-acceptance, gray line represents those with high psychological capital but low self-acceptance, tangerine line represents low psychological capital but high self-acceptance, and purple line represents low psychological capital and self-acceptance. Individuals with high psychological capital and self-acceptance show the best health status regardless of depressive tendency. High depressive tendency worsens health status in comparison to low one at the same level of psychological capital and self-acceptance.

**Table 1 behavsci-13-00552-t001:** Differences in self-rated health, depressive tendency, psychological capital, and self-acceptance among demographics and lifestyles.

Variable	Self-Rated Health		Depressive Tendency		Psychological Capital		Self-Acceptance	
	N	$\bar{x}\pm s$	N	$\bar{x}\pm s$	N	$\bar{x}\pm s$	N	$\bar{x}\pm s$
Sex								
Male	507	333.23 ± 56.81	505	17.59 ± 9.75	507	107.22 ± 20.27	507	41.68 ± 5.78
Female	931	335.11 ± 49.79	929	15.82 ± 8.99	931	105.38 ± 17.68	931	41.24 ± 6.08
*t*		−0.626		3.388 **		1.725		1.321
Having sibling								
No	201	331.83 ± 52.12	200	17.00 ± 9.62	201	108.63 ± 17.89	201	41.68 ± 5.61
Yes	1237	334.87 ± 52.40	1234	16.35 ± 9.25	1237	105.60 ± 18.74	1237	41.35 ± 6.04
*t*		−0.766		0.906		2.134 *		0.716
Exercise (times per week)								
>5	114	342.49 ± 59.95	114	16.02 ± 10.29	114	111.69 ± 22.78	114	42.99 ± 6.93
3–4	199	343.59 ± 53.71	198	15.55 ± 9.67	199	111.40 ± 18.31	199	42.73 ± 5.52
1–2	437	341.90 ± 49.12	436	15.76 ± 9.18	437	107.32 ± 17.86	437	42.05 ± 5.72
1	545	329.13 ± 51.27	543	17.09 ± 9.19	545	103.80 ± 17.26	545	40.61 ± 5.77
0	143	312.76 ± 49.58	143	17.65 ± 8.57	143	98.59 ± 19.31	143	39.27 ± 6.34
*F*		12.319 ***		2.366		15.509 ***		13.129 ***
Online surfing (hours)								
<3	642	340.47 ± 53.00	641	16.09 ± 9.54	642	108.48 ± 19.31	642	42.46 ± 5.97
3~6	600	331.66 ± 50.27	599	16.40 ± 8.85	600	105.12 ± 17.28	600	40.81 ± 5.78
>6	196	323.24 ± 52.36	194	17.74 ± 9.80	196	100.79 ± 19.24	196	39.70 ± 6.02
*F*		9.710 ***		2.350		14.269 ***		21.651 ***

Note: * *p* < 0.05, ** *p* < 0.01, *** *p* < 0.001.

**Table 2 behavsci-13-00552-t002:** Pearson correlations between self-rated health, depressive tendency, psychological capital, and self-acceptance.

Variable	*N*	*M*	*SD*	Self-Rated Health	Depressive Tendency	Psychological Capital
Self-rated Health	1438	334.45	52.36	1		
Depressive tendency	1434	16.44	9.30	−0.63 **	1	
Psychological capital	1438	106.03	18.65	0.56 **	−0.46 **	1
Self-acceptance	1438	41.40	5.98	0.52 **	−0.56 **	0.51 **

** *p* < 0.01.

**Table 3 behavsci-13-00552-t003:** Regression analysis of depressive tendency, psychological capital, self-acceptance, and self-rated health.

	Phase 1 (n = 367)		Phase 2 (n = 1071)	
Variable	*β*	*p* Value	*β*	*p* Value
	Model 1			
Depressive tendency	−0.53	<0.001	−0.40	<0.001
Psychological capital	0.27	<0.001	0.30	<0.001
Self-acceptance	0.11	0.011	0.13	<0.001
Adjusted R^2^	0.626	0.472		
ΔR^2^	0.629	<0.001	0.473	<0.001
	Model 2			
Depressive tendency	−0.51	<0.001	−0.40	<0.001
Psychological capital	0.28	<0.001	0.30	<0.001
Self-acceptance	0.11	0.016	0.13	<0.001
Depressive tendency × psychological capital	0.04	>0.05	−0.02	>0.05
Depressive tendency × self-acceptance	0.08	>0.05	0.02	>0.05
Psychological capital × self-acceptance	0.09	>0.05	0.02	>0.05
Adjusted R^2^	0.628	0.471		
ΔR^2^	0.006	>0.05	0.001	>0.05
	Model 3			
Depressive tendency	−0.54	<0.001	−0.41	<0.001
Psychological capital	0.31	<0.001	0.32	<0.001
Self-acceptance	0.16	0.001	0.16	<0.001
Depressive tendency × psychological capital	0.08	>0.05	−0.02	>0.05
Depressive tendency × self-acceptance	0.08	>0.05	0.03	>0.05
Psychological capital × self-acceptance	0.13	0.014	0.02	>0.05
Depressive tendency× psychological capital × self-acceptance	0.13	0.013	0.07	0.025
Adjusted R^2^	0.634	0.473		
ΔR^2^	0.006	0.013	0.002	0.025

**Table 4 behavsci-13-00552-t004:** Mediating effect of psychological capital and self-acceptance on depressive tendency and self-rated health.

Mediating Variable	Depressive Tendency (a)	Self-Rated Health (b)	Total Effect (c)	Direct Effect (c’)	Indirect Effect (ab) (95%CI)	Percentage (ab/c) %
Psychological capital	−0.93 ***	0.95 ***	−3.58 ***	−2.70 ***	−0.88 (−1.05, −0.73)	24.7
Self-acceptance	−0.36 ***	2.05 ***	−3.58 ***	−2.83 ***	−0.74 (−0.90, −0.60)	20.6

Note: *** *p* < 0.001; a, the effect of the independent variable on mediating variable; b, the effect of mediating variable on self-rated health; c, the total effect of the independent variable on self-rated health; c’, the direct effect of the independent variable on self-rated health after the introduction of mediating variable; a × b, the mediating effect of mediating variable between depressive tendency and self-rated health.

## Data Availability

The dataset supporting the conclusions of this article can be shared with the Corresponding author by email.

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
