# Peer review of "Psychological Capital and Self-Acceptance Modified the Association of Depressive Tendency with Self-Rated Health of College Students in China during the COVID-19 Pandemic"

_behavsci, 2023, doi:10.3390/bs13070552_

Round 1
Reviewer 1 Report
I read with great interest the manuscript entitled 'Psychological capital and self-acceptance modified the association of depressive tendency with self-rated health status of college students in China during the pandemic of COVID-19'. I think it is a well-written article, worthy of investigation in a field where relatively little literature exists.
The abstract explains the purpose of the paper directly and clearly.
The introduction and discussion are well articulated. The article therefore presents a nice overview of the relevant literature on this topic and adds significant theoretical and practical implications, and I enjoyed reading it.
The procedure is clearly argued as are the results, which are summarised in meaningful tables for the readability of the work. In the discussion, the results are interpreted on the basis of the relevant literature in a comprehensive manner. Limitations and future implications are also comprehensively stated.
While I do not have any major observations, I would only suggest the possibility of including some recent and valuable contributions on this topic in the introduction (see e.g. Prestia et al.,The impact of the COVID-19 pandemic on patients with OCD: Effects of contamination symptoms and remission state before the quarantine in a preliminary naturalistic study (2020) ) and to expand the explanation of psychological capital as it concerns key aspects of psychology that can hardly be summarised in a few words including resilience. Within the limits this could be indicated, namely that it would be interesting to implement the study using individual aspects of psychology such as resilience to understand possible differences or similarities in the results. Finally, I suggest removing references to statistical data in the abstract as I think it can be easily read and understood that way.
Author Response
Dear Editor
Thank you for your time and consideration of our work. We also want to extend our appreciation to the reviewers for their critical comments and thoughtful suggestions, which have substantially improved our work. Following the suggestions, we have carefully revised our work, which are underscored in the revised manuscript. We also addressed point-by-point the concerns raised by the reviewers as follows.
Thank you again for consideration of our work.
Yongcheng Yao
Point-by-point response:
Reviewer 1
I read with great interest the manuscript entitled 'Psychological capital and self-acceptance modified the association of depressive tendency with self-rated health status of college students in China during the pandemic of COVID-19'. I think it is a well-written article, worthy of investigation in a field where relatively little literature exists.
The abstract explains the purpose of the paper directly and clearly.
The introduction and discussion are well articulated. The article therefore presents a nice overview of the relevant literature on this topic and adds significant theoretical and practical implications, and I enjoyed reading it.
The procedure is clearly argued as are the results, which are summarised in meaningful tables for the readability of the work. In the discussion, the results are interpreted on the basis of the relevant literature in a comprehensive manner. Limitations and future implications are also comprehensively stated.
Answer: Thank the reviewer for his/her positive feedback.
While I do not have any major observations, I would only suggest the possibility of including some recent and valuable contributions on this topic in the introduction (see e.g. Prestia et al.,The impact of the COVID-19 pandemic on patients with OCD: Effects of contamination symptoms and remission state before the quarantine in a preliminary naturalistic study (2020) )
Answer: Thanks for the excellent reference. We added it in the introduction appropriately on page 2 line 57-58 as “it also led to a variety of psychological adverse effects, e.g., anxiety, fear, grief, depression[7] and other serious health problems (e.g., obsessive-compulsive disorder (OCD) (Davide P, PMID: 32535508), due to fewer opportunities to engage with peers and/or excessive online surfing for communication, fear of future and change in academics”
and to expand the explanation of psychological capital as it concerns key aspects of psychology that can hardly be summarised in a few words including resilience.
Answer: Following the suggestion, we expand the explanation of four aspects of psychological capital by adding the sentences as “Self-efficacy makes an individual take effort with confidence to succeed at challenging tasks. When Facing difficulties, one can persevere toward goals with a positive attribution and expectation, redirect paths to goals in order to achieve (optimism and hope). Even beset by problems and failure, one can bounce back from the adversity to attain success (resilience)” on page 3 line 125-130.
Within the limits this could be indicated, namely that it would be interesting to implement the study using individual aspects of psychology such as resilience to understand possible differences or similarities in the results.
Answer: As suggested, we added one sentence as “It would also be interesting to investigate the association of individual aspects of psychological capital each such as resilience with depression tendency in college students to understand possible differences or similarity among the four different aspects in psychological capital during such humanity tragedies” in the paragraph of limitation on page 11 line 169-173.
Finally, I suggest removing references to statistical data in the abstract as I think it can be easily read and understood that way.
Answer: Thank the reviewer for the suggestion. We guess that the reviewer suggested to move the references cited in the statistical methods to the abstract. If we understand it is correct, we think reference citations are not allowed in the abstract in this journal.
Reviewer 2 Report
Dear Author -The study "Psychological capital and self-acceptance modified the association of depressive tendency with self-rated health status of college students in China during the pandemic of COVID-19" is a quantitative survey on undergraduate students that aims to understand students' mental health in relation to psychological capital (PC) and self acceptance (SA) in the context of the COVID pandemic. The study offers information about student mental health and its relationship to PC and SA. I have enjoyed reading this manuscript. The result section is very impressive.
The following can be added to the manuscript to make it better. In PDF, some of the comments are already embedded.
1. The need for the study is not sufficiently stated in the introduction section.
2. The study's introduction section is missing pertinent citations that could have been used to support its focus on undergraduate students in China or Covid 19. This may be included.
3. The variables being investigated are not clearly defined or conceptually linked. The study's justification is missing.
4. Ethics and the specific demographic and background data collected are not covered in the method section. Sex-related data is gathered.
5. The first section of the result presented as an association rather than a comparison with demographic and background data.
6. The results section is comprehensive and contains the majority of the data required to support the study.
7. The sequence of explanations in the discussion section, which should have begun with demographics, background information, and key variables, is lacking. The findings of the current study are poorly explained and not well supported with the existing researches in this section.

Author Response
Dear Editor
Thank you for your time and consideration of our work. We also want to extend our appreciation to the reviewers for their critical comments and thoughtful suggestions, which have substantially improved our work. Following the suggestions, we have carefully revised our work, which are underscored in the revised manuscript. We also addressed point-by-point the concerns raised by the reviewers as follows.
Thank you again for consideration of our work.
Yongcheng Yao
Point-by-point response:
Reviewer 2
Dear Author -The study "Psychological capital and self-acceptance modified the association of depressive tendency with self-rated health status of college students in China during the pandemic of COVID-19" is a quantitative survey on undergraduate students that aims to understand students' mental health in relation to psychological capital (PC) and self acceptance (SA) in the context of the COVID pandemic. The study offers information about student mental health and its relationship to PC and SA. I have enjoyed reading this manuscript. The result section is very impressive.
Answer: Thank the reviewer for his/her time and positive feedback.
The following can be added to the manuscript to make it better. In PDF, some of the comments are already embedded.
- The need for the study is not sufficiently stated in the introduction section.
Answer: line 55-61 on page 2: we agree with the reviewer, isolation is not a sole factor in causing mental health. We revised the sentence as “ This physical isolation measurement significantly flattens the disease curve. However, it also led to a variety of psychological adverse effects, e.g., anxiety, fear, grief, depression[7] and other serious health problems (e.g., obsessive-compulsive disorder (OCD) (Davide P, PMID: 32535508), due to fewer opportunities to engage with peers and/or excessive online surfing for communication, fear of future and change in academics”.
We also re-organized the paragraphs in the introduction as suggested.
- The study's introduction section is missing pertinent citations that could have been used to support its focus on undergraduate students in China or Covid 19. This may be included.
Answer: Following the suggestion, we added two more sentences with related references as “A recent study reported that 21.1% undergraduate and postgraduate college students (aging 18 years and older) from 152 countries reported depression (Ellakany P, 2023, PMID: 37264389). In Shanghai, depressive symptoms dramatically increased in college students with the introduction of the quarantine in periods 2 and 3 compared to the period 1 (Ma D, 2023, PMID: 37213647).” On page 2 line 70-74.
- The variables being investigated are not clearly defined or conceptually linked. The study's justification is missing.
Answer: We re-organized the introduction to make it conceptually linked. The purposes of this study were to evaluate the mental health status in Chinese college students and to investigate the associations between depression, psychological capital, self-acceptance and self-rated health during the COVID-19 pandemic. In the introduction, we included enough evidence linking COVID-19, depression/self-rated health self-acceptance, and psychological capital in college students, although we could not include all references in the introduction. We made some revisions to make it clearly.
- 4. Ethics and the specific demographic and background data collected are not covered in the method section. Sex-related data is gathered.
Answer: In the method section, we gave some examples such as “sex, age, sibling, exercise (above half an hour each time) and the Internet surfing duration every day” on page 4 line 179-180.
- The first section of the result presented as an association rather than a comparison with demographic and background data.
Answer: Following the suggestion, we revised the subtitle 1 as “Differences in self-rated health, depressive tendency, psychological capital and self-acceptance among demographics and lifestyles” (on page 5) and in Table 1 (on page 6)
- The results section is comprehensive and contains the majority of the data required to support the study.
Answer: Thank the reviewer for positive comment on the result section.
- The sequence of explanations in the discussion section, which should have begun with demographics, background information, and key variables, is lacking. The findings of the current study are poorly explained and not well supported with the existing researches in this section.
Answer: Thank the reviewer for your kind suggestion on the sequence of explanations in the discussion section. We preferred to discussion the major findings first on the association between depression, self acceptance, psychological capital and their interaction. Then we discussed the association between demographics and other variables with depression tendency. We agree with the reviewer on the association of sibling and depression during the pandemic, which was beyond the main purposes.
We thank the reviewer for suggestion in the conclusion. Following the suggestion, we added the sentences as “Although the end of COVID-19 pandemic has been announced, post-effects on mental health may not vanish immediately. In addition, humanity may time to time suffer from such pandemic in the future, and such studies could have useful insight in these contexts” on page 12 lines 185-189.
Round 2
Reviewer 2 Report
Dear Author,
I appreciate the manuscript's changes.
1. The link is much better, and the introduction section has been improved.
2. It is appropriate to include differences in line 240. I'm not sure if the line spacing continues to match the rest of the section after line 241. Please double-check it.
3. In line 429, the word humanity tragedy conveys a different meaning. It is not appropriate to introduce it abruptly here because it is not used anywhere in the essay.
4. The additional lines from 441 to 445 are excellent.
I still recommend below two points for revisions.
1. It is possible to add a thematic link to the paper in intro section.
2. The current draft of the manuscript does not yet incorporate point 7.
The discussion section's explanations don't follow the proper order, which would have started with background data, demographics, and important variables. The results of the current study are not adequately explained or supported by the previous studies in this section.
Author Response
I appreciate the manuscript's changes.
- The link is much better, and the introduction section has been improve d.
Answer: Thanks for positive feedback
- It is appropriate to include differences in line 240. I'm not sure if the line spacing continues to match the rest of the section after line 241. Please double-check it.
Answer: Thanks, it was adjusted accordingly.
3.In line 429, the word humanity tragedy conveys a different meaning. It is not appropriate to introduce it abruptly here because it is not used anywhere in the essay.
Answer: We replaced it as “the COVID-19 pandemic” in line 429
- The additional lines from 441 to 445 are excellent.
Answer: thanks
I still recommend below two points for revisions.
- It is possible to add a thematic link to the paper in intro section.
Answer: Following the suggestion, we added it accordingly, if we understand it correctly your meaning of “thematic link to the paper”. We thought this information usually should be added in the editor office when published.
- The current draft of the manuscript does not yet incorporate point 7.
Answer: Following the suggestion, we ordered the sequence of explanation in the discussion, hopefully it works.